# The global climate response to High-Latitude Low-Altitude Stratospheric Aerosol Injection (HiLLA-SAI)

Alistair Duffey<sup>1</sup>, Walker Lee<sup>2</sup>, Lauren Wheeler<sup>3</sup>, Peter Irvine<sup>4</sup>, Benjamin Wagman<sup>5</sup>, Matthew Henry<sup>6</sup>, Daniele Visioni<sup>7</sup>, Michel Tsamados<sup>1</sup>, and Douglas MacMartin<sup>8</sup>

Correspondence: Alistair Duffey (alistair.duffey.21@ucl.ac.uk)

Abstract. High-latitude low-altitude (HiLLA) Stratospheric Aerosol Injections (SAI) would face fewer logistical barriers than high-altitude low-latitude SAI, because it could use repurposed existing large aircraft for deployment. However, relative to high-altitude SAI, it is expected to have reduced global cooling efficiency, and the more polar forcing profile and reduced tropical stratospheric heating would result in many differences in the surface climate response. Here, we present the first multi-model simulations of HiLLA-SAI, in UKESM1, CESM2-WACCM and E3SMv3. Using these simulations, we assess the global climate response to HiLLA-SAI, and the sensitivity to the latitude, altitude (13 km versus 15 km), seasonality and longitude of injections. For seasonal injections at 60°N/S and 13 km, all models show similar global cooling efficiency, of around 0.6°C per 12 Mt SO<sub>2</sub> per year, 40-53% of the equivalent cooling efficiency for 21 km injection in the tropics. Raising the injection height to 15 km increases this global cooling efficiency by around half, to 63-70% of the high altitude tropical case. The effects of HiLLA-SAI are more polar focused than other SAI strategies, particularly for the 13 km injection case, and large changes in sea-ice in both hemispheres, high-latitude precipitation and the polar seasonal cycle are shown. Nevertheless, our results highlight that HiLLA-SAI would still be a global intervention. For 13 km inject, tropical cooling per unit global cooling is 61-75% of the rate under greenhouse-gas forced warming, and is larger in the 15 km case. Precipitation changes and sulfur deposition are also found at all latitudes. Overall, our results highlight the importance of further study into HiLLA-SAI strategies, which these simulations suggest could be a viable early-stage SAI deployment strategy, with global, not just polar, impacts.

<sup>&</sup>lt;sup>1</sup>Department of Earth Sciences, University College London, London, UK

<sup>&</sup>lt;sup>2</sup>Climate & Global Dynamics Division, National Science Foundation's National Center for Atmospheric Research, Boulder, CO. USA

<sup>&</sup>lt;sup>3</sup>Atmospheric Sciences Department, Sandia National Laboratories, Albuquerque, NM, USA

<sup>&</sup>lt;sup>4</sup>Department of Geophysical Sciences, University of Chicago, Chicago, Illinois, 60637, United States

<sup>&</sup>lt;sup>5</sup>Earth Systems Analysis Department, Sandia National Laboratories, Albuquerque, NM, USA

<sup>&</sup>lt;sup>6</sup>Department of Mathematics, University of Exeter, Exeter, UK

<sup>&</sup>lt;sup>7</sup>Department of Earth and Atmospheric Sciences, Cornell University, Ithaca, NY, USA

<sup>&</sup>lt;sup>8</sup>Sibley School of Mechanical and Aerospace Engineering, Cornell University, Ithaca, NY, USA

35

## 1 Introduction

Stratospheric aerosol injection (SAI) is a proposed means of reducing global temperature rise and associated changes in climate by adding aerosols or their precursors to the stratosphere (Crutzen, 2006; NASEM, 2021; UNEP, 2023). These aerosols would scatter a small fraction of incoming sunlight, as occurs after some volcanic eruptions (Robock, 2000, 2013), producing a negative radiative forcing. Most scenarios of SAI envision deployment at altitudes greater than 20 km (Richter et al., 2022; Tilmes et al., 2018), well above the altitude at which existing large aircraft could deploy sufficient quantities of SO<sub>2</sub>. However, the tropopause is lower in the polar regions, making it plausible that deployment in the mid- to high-latitudes using existing commercial aircraft at altitudes near their limit (i.e. 13 - 15 km) could achieve sufficient cooling efficiency to be a viable climate intervention (Duffey et al., 2025a). Such low-altitude deployment has been suggested as a potential strategy for the early years of a future deployment scenario (Wheeler et al., 2025). The reduced barriers to acquiring a fleet capable of this deployment strategy could also imply an increase in the number of actors able to participate in SAI and an earlier possible start date (Smith et al., 2024; Smith, 2024).

High-latitude SAI deployment has also been proposed as a means of reducing the rapid climate change ongoing in the Arctic (Robock et al., 2008; Lee et al., 2023a), which is warming three to four times faster than the global average (Rantanen et al., 2022) and is on track to be sea-ice free in summer by mid-century (Notz and Stroeve, 2018; Notz and Community, 2020; Bonan et al., 2021). Preventing or slowing the passing of cryospheric tipping points (e.g. Armstrong McKay et al., 2022), has also been proposed as a target of polar SAI deployments (Smith et al., 2024; Futerman et al., 2025), and in the Antarctic, SAI has been shown to reduce the loss of land ice (Goddard et al., 2023).

Motivated by these implications, several recent studies have assessed such 'high-latitude low-altitude' (HiLLA) scenarios in earth system models (Lee et al., 2021, 2023a; Wheeler et al., 2025; Duffey et al., 2025a). Here, we build on these studies to present the first multi-model comparison of simulations of HiLLA-SAI, using three earth system models - the UK Earth System Model (UKESM1), the Community Earth System Model (CESM2-WACCM), and the Energy Exascale Earth System Model (E3SMv3) - to investigate the global and polar climate impacts of HiLLA-SAI and their sensitivity to the altitude, longitude and seasonality of injection.

Deploying SAI in polar or sub-polar latitudes, and at lower altitudes, is expected to reduce the global cooling efficiency, defined as the magnitude of near-surface air temperature change per unit injected SO<sub>2</sub>, for two main reasons. First, the combination of reduced distance in the stratosphere through which the aerosols sediment and a less favourable background circulation decreases the aerosol lifetime and consequently the steady-state burden under a given injection rate (Niemeier et al., 2011; Tilmes et al., 2017; Lee et al., 2023b); In UKESM1, HiLLA-SAI strategies show lifetimes of 2-3 months, as compared to over 8 months for injection at 30°N/S and 20km (Duffey et al., 2025a). Qualitatively similar lifetimes are found by Lee et al. (2023a) for 15 km injection at 60°N in CESM2-WACCM, and in the earlier study of Robock et al. (2008), who simulate injection at 68°N, 10-15 km altitude. Given the short lifetime and the strong seasonality of insolation at high-latitudes, HiLLA strategies would necessarily be seasonal, with injection only during local spring/summer (Lee et al., 2021). Second, higher latitude injection would produce a more polar distribution of aerosols, such that they occur over regions of lower insolation

(except in local mid-summer) (North et al., 1981), high cloud cover (Intrieri et al., 2002), and high surface albedo, which all contribute to a reduced forcing per unit aerosol optical depth (e.g. Duffey et al., 2025a).

On the other hand, there are compensating effects which would oppose the decrease in efficiency for HiLLA strategies. As with the Arctic amplification of greenhouse-gas forced warming (e.g. Serreze and Francis, 2006; Pithan and Mauritsen, 2014; Taylor et al., 2022), local feedbacks also drive greater change in Arctic temperature for a given radiative forcing under SAI. This leads to greater absolute cooling in the Arctic than elsewhere under even low latitude injection in both UKESM1 (Henry et al., 2024) and CESM2-WACCM (Zhang et al., 2024). As a result, global mean temperature change is more susceptible to a given fractional restoration of Arctic than tropical climate (if counted over the same surface area of the planet). Additionally, aerosols with longer lifetimes are also larger on average, having more time to grow, and so can be less optically efficient (Kleinschmitt et al., 2018), although this effect is likely outweighed by the extra burden from increased lifetime (Lee et al., 2023b).

Beyond the overall global cooling efficiency, HiLLA-SAI would result in many important differences relative to the lower latitude and higher altitude SAI strategies which have been more commonly investigated. Drivers of these differences include the more polar and seasonal pattern of radiative forcing, and the strongly reduced tropical stratospheric heating (Bednarz et al., 2023a). Additionally, we expect a smaller tropical lower stratospheric water vapour increase due to a more polar aerosol burden under HiLLA-SAI, and greater lower-stratospheric ozone depletion due to the higher sulfate aerosol concentrations in the high latitudes (Bednarz et al., 2023a). In the context of volcanic eruptions, the natural analogue for SAI, low and high latitude eruptions are associated with different surface climate responses (Robock, 2000; Schneider et al., 2009; Sjolte et al., 2021). A strengthened polar vortex and positive North Atlantic Oscillation signal is associated with tropical stratospheric heating caused by low-latitude eruptions (Robock and Mao, 1992; Stenchikov et al., 2002; Zambri and Robock, 2016), but this effect is reduced (Schneider et al., 2009) or reversed (Sjolte et al., 2021) under high-latitude eruptions.

In this study, our simulations use an idealized injection strategy, injecting a constant mass of  $SO_2$  each year at a single location in each hemisphere into the sub-polar lower stratosphere. This strategy is not designed to represent a plausible policy scenario. Instead, here, as with several previous intermodel comparisons of SAI (e.g. Kravitz et al., 2013; Visioni et al., 2023), we use a simple scenario to explore model sensitivities and inform future simulations of more complex HiLLA scenarios which could vary injections to target particular climate outcomes. We assess, first, the temperature change under HiLLA-SAI in the three models, globally and regionally. Second, we assess the aerosol distributions and optical depth (AOD), and the variation of these with the precise injection strategy. Third, we make a more detailed assessment of the response in the polar regions, including the seasonality of Arctic cooling and the sea-ice impacts. Finally, we briefly investigate several impacts which might be expected to differ between HiLLA and conventional high-altitude SAI: sulfate deposition, stratospheric ozone, and changes in tropical precipitation. Resolving the various uncertainties associated with the processes described here, to allow a comprehensive statement of the trade-offs between HiLLA and other SAI strategies, will require substantial further research. The results we describe have informed the design of a new GeoMIP experiment, G6-1.5K-HiLLA (Visioni et al., 2025), which will support this community research effort.

## 85 2 Methods

## 2.1 Earth System Models

We use an ensemble of simulations in three earth system models; UKESM1, CESM2-WACCM and E3SMv3. The version of UKESM1 used here has been used extensively to model SAI in recent years (Jones, 2019; Jones et al., 2021; Henry et al., 2023; Visioni et al., 2023; Bednarz et al., 2023b). The model uses the coupled climate model HadGEM3-GC3.1 as the physical core (Kuhlbrodt et al., 2018), coupled to various earth system components including terrestrial and ocean biochemistry and dynamic vegetation (Sellar et al., 2019). There is a unified tropospheric and stratospheric chemistry model, UKCA (Archibald et al., 2020). The aerosol model, GLOMAP (Mann et al., 2010) is used with five log-normal modes (Mulcahy et al., 2018, 2020). The atmosphere has 85 vertical levels up to 85km, and a 1.875° longitude by 1.25° latitude resolution.

The Community Earth System Model, version 2 (CESM2; Danabasoglu et al., 2020) is a state-of-the-art, fully-coupled Earth system model with prognostic atmosphere, ocean, land, sea ice, land ice, river runoff, and wave components developed by the U.S. National Science Foundation's National Center for Atmospheric Research. The model is run at a horizontal resolution of 1.25° longitude x 0.9° latitude. The atmospheric component used for these simulations is the Whole Atmosphere Community Climate Model (WACCM6; Gettelman et al., 2019), which uses a finite-volume dynamical core and features a higher model top (70 vertical levels, up to approximately 140 km) and more complex chemistry and physics representation than the Community Atmosphere Model. This configuration, CESM2(WACCM6), hereafter CESM2-WACCM, contributed to Phase 6 of the Climate Model Intercomparison Project (CMIP6; Eyring et al., 2016) and has been extensively used to model SAI (e.g. Tilmes et al., 2020; Richter et al., 2022). The configuration uses version 4 of the Modal Aerosol Module (MAM4; Liu et al., 2016), which features prognostic aerosol representation and represents sulfate aerosols in Aitken, accumulation, and coarse modes. The ocean component is version 2 of the Parallel Ocean Program (POP2; Smith et al., 2010; Danabasoglu et al., 2012, 2020), and the land component is version 5 of the Community Land Model (CLM5; Lawrence et al., 2019).

The Energy Exascale Earth System Model version 3 (E3SMv3) is developed by the U.S. Department of Energy (DOE) and is a fully coupled earth system model with five components for the atmosphere, land, river, ocean, and sea ice (Golaz et al., 2025). This is the first use of E3SMv3 for simulations of SAI. Previous versions of E3SMv2 and E3SMv2-PA (Brown et al., 2024) have been validated against the Mount Pinatubo eruption (Brown et al., 2024; Hu et al., 2025) and used for simulations of SAI (Wheeler et al., 2025). E3SMv3 is run with three different horizontal grids for the atmosphere, ocean and sea ice, and land and river components. For the simulations in this paper, the 'ne30pg2\_r05\_IcoswISC30E3r5' grid is used. The 'ne30pg2\_r05\_IcoswISC30E3r5' grid includes a cubed-sphere grid with a spacing of ~110 and ~165 km for the atmosphere dynamics and physics grids, an unstructured horizontal grid with 20-30 km resolution for the ocean and sea-ice components, and a 0.5° regular latitude-longitude grid for the land and river components (Golaz et al., 2025). The atmosphere component is the E3SMv3 atmosphere model (EAM; Xie et al., 2025), which has improved vertical resolution in the lower stratosphere with 80 vertical levels (up to approximately 60 km) relative to E3SMv2 (Golaz et al., 2022). Prognostic aerosols are simulated with the Modal Aerosol Module with Prognostic Stratospheric Aerosol (MAM5-SPA) which includes a prognostic sulfate aerosol

**Figure 1.** Tropopause altitude in UKESM1, CESM2-WACCM, E3SMv3 and the ERA5 reanalysis, as well as the locations of the HiLLA-13 and HiLLA-15 simulations, and 3-year sensitivities. Tropopause values shown are the time means over the final 25 years of the historical scenario (1991-2015) for for UKESM1 and CESM2-WACCM and the same period for ERA5. For E3SMv3, which had no historical scenario available, the time period is 2035-2050. Shaded regions show the minimum-maximum range of the monthly tropopause height over the four months during the 25-year period. The tropopause definition for models is the World Meteorological Organization (WMO) 1st thermal tropopause, and for ERA5 is the dynamical.

scheme for stratospheric coarse sulfate particles into MAM4 (Hu et al., 2025). The ocean component is MPAS-Ocean ((Smith et al., 2025; Golaz et al., 2025) and the land component is the E3SMv3 land model (ELMv3; Golaz et al., 2025).

## 120 2.2 Simulations

125

The specifications for all simulations are shown in Table 1. The injection locations in latitude/altitude space are shown in Figure 1, along with the tropopause position as modelled in UKESM1, CESM2-WACCM, E3SMv3, and in the ECMWF Reanalysis version 5 (ERA5) reanalysis (Hersbach et al., 2020), with zonal mean tropopause height data taken from Hoffmann and Spang (2022). We designed a sensitivity suite to test the response to changes in injection strategy for HiLLA scenarios. Across these simulations we vary injection altitude, latitude, seasonality, and longitude. All simulations of SAI are run under the SSP2-4.5 pathway, branching and initiating SAI in 2035 and all inject a fixed magnitude of 12 Tg SO<sub>2</sub> per year in total following the same protocol used for low-latitude high-altitude injections in Visioni et al. (2023). The rate of injection varies across the

130

simulations to keep the total annual injection constant, given the varying injection period between three and five months in each hemisphere per year.

There are two central HiLLA scenarios which serve as the baseline SAI scenarios against which we compare the shorter sensitivity simulations: HiLLA-13 and HiLLA-15, each of which run for 35 years. These inject SO<sub>2</sub> for 4 months in the Northern Hemisphere spring/early-summer (March-June) and 4 months in the Southern Hemisphere (September-December). HiLLA-13 and HiLLA-15 inject at 13 km and 15 km, respectively, at 60° North and South. The '13 km' and '15 km' heights by which we label the simulations are nominal only. In UKESM1, which operates on hybrid-height vertical coordinates, the vertical resolution at 13-15 km is approximately 600 m, and the SO<sub>2</sub> is injected into grid boxes injected which have centres at 12.9 and 15.4 km, respectively. In CESM2-WACCM, injections are into grid cells bounded by pressure surfaces at 12.9-13.0 km for HiLLA-13, and at 14.7-14.9 km for HiLLA-15. In E3SMv3, the injection grid cells are bounded at 12.5-13.5 and 14.5-15.5 km altitudes, with centers exactly at the nominal altitudes.

In addition to the two central HiLLA scenarios, we also simulated nine short (3-year duration) simulations in each model. Across these nine simulations, we vary (independently) the injection latitude, altitude, seasonality (within the spring/early summer strategy previously shown to be significantly more effective (Lee et al., 2021)), and longitude. We do not attempt to fully explore this four-dimensional injection scenario space, but instead vary each feature independently, relative to our central HiLLA cases, to show a first-order estimate of the sensitivity to each.

We also use the ARISE-SAI-1.5 (hereafter "ARISE") simulations, in CESM2-WACCM (Richter et al., 2022) and UKESM1 (Henry et al., 2023), as a comparison representing low-latitude high-altitude (i.e. "conventional") SAI. These simulations use injection of SO<sub>2</sub> at 15°N/S and 30°N/S, and at approximately 21km, with the injection at each latitude controlled by a feedback algorithm targeting the meridional pattern of temperature response, as well as the global mean temperature. See Richter et al. (2022) for a full description of the ARISE scenario. The comparison of our HiLLA scenarios against ARISE is imperfect, since here we do not control the injection to achieve the same outcome state. While we reduce the impact of this difference by scaling effects by unit cooling or injection, it is still an important limitation to our study. The upcoming G6-1.5K-HiLLA simulations will allow for analyses not subject to this limitation, since they will be directly comparable against G6-1.5K-SAI simulations which will achieve the same global temperature target via an alternative strategy (Visioni et al., 2024, 2025).

#### 3 Results

## 3.1 Temperature response

The global, polar, and tropical near-surface air temperature evolution under the background SSP2-4.5 and the central HiLLA-13 and HiLLA-15 scenarios, in each model, are shown in Figure 2. The HiLLA-13 case produces approximately 0.6°C of global mean cooling (Figure 3), which is reached after around 10 years of deployment (Figure A1) under the constant injection magnitude used in this scenario. More precisely, UKESM1, CESM2-WACCM and E3SMv3 see 0.58°C, 0.70°C and 0.65°C of global mean cooling over the final 20 years of the simulation (2050-2069), respectively. Raising the altitude of injection to 15 km increases the global cooling efficiency by 60%, 34% and 62% in UKESM1, CESM2-WACCM and E3SMv3, respectively.

**Table 1.** Overview of the simulations used in this study. The central cases were run for 35 years, while sensitivity cases were run for 3 years to assess the impact of varying parameters. Month ranges are inclusive in all cases, i.e. MAMJ/SOND means injection in for four months March-June in the Northern Hemisphere, and four months September-December in the Southern Hemisphere. Due to an error in the injection settings, E3SMv3 does not have output for the 2nd seasonality sensitivity simulation, with injection in MAMJJ/SONDJ.

| Label         | Altitude (km) | Latitude (°N/S) | Months                           | Longitude ( $^{\circ}$ ) | Length (years) |
|---------------|---------------|-----------------|----------------------------------|--------------------------|----------------|
| Central cas   | es            |                 |                                  |                          |                |
| HiLLA-13      | 13            | 60              | Mar-Jun (MAMJ)/ Sep-Dec (SOND)   | 180                      | 35             |
| HiLLA-15      | 15            | 60              | Mar-Jun (MAMJ)/ Sep-Dec (SOND)   | 180                      | 35             |
| Sensitivities | 5             |                 |                                  |                          |                |
| Latitude      |               |                 |                                  |                          |                |
|               | 13            | 70              | Mar-Jun (MAMJ)/ Sep-Dec (SOND)   | 180                      | 3              |
|               | 13            | 50              | Mar-Jun (MAMJ)/ Sep-Dec (SOND)   | 180                      | 3              |
| Altitude      |               |                 |                                  |                          |                |
|               | 15            | 70              | Mar-Jun (MAMJ)/ Sep-Dec (SOND)   | 180                      | 3              |
|               | 15            | 50              | Mar-Jun (MAMJ)/ Sep-Dec (SOND)   | 180                      | 3              |
| Seasonality   |               |                 |                                  |                          |                |
|               | 13            | 60              | Mar-May (MAM)/ Sep-Nov (SON)     | 180                      | 3              |
|               | 13            | 60              | Mar-Jul (MAMJJ)/ Sep-Jan (SONDJ) | 180                      | 3              |
|               | 13            | 60              | Feb-Jun (FMAMJ)/ Aug-Dec (ASOND) | 180                      | 3              |
| Longitude     |               |                 |                                  |                          |                |
|               | 13            | 60              | Mar-Jun (MAMJ)/ Sep-Dec (SOND)   | 0                        | 3              |
|               | 15            | 60              | Mar-Jun (MAMJ)/ Sep-Dec (SOND)   | 0                        | 3              |

The HiLLA-SAI cases produce a polar focused cooling profile (Figure 4). The feedbacks which drive Arctic amplification mean that even low-latitude SAI produces more absolute temperature change in the Arctic than the global mean (Henry et al., 2024). However, the greater Arctic cooling is particularly strong for HiLLA scenarios, which show 3.8 (UKESM1), 2.7 (CESM2-WACCM), and 2.4 (E3SMv3) times more Arctic (>66°N) than global mean cooling, for the 13 km injection case. Part of the inter-model difference in this Arctic to Global cooling ratio can be explained by the relative Arctic amplification ratios of the models (3.2, 1.8, and 2.6) under SSP2-4.5. However, this is not the complete explanation, since CESM2-WACCM has a smaller Arctic amplification ratio than E3SMv3 under warming, but a higher Arctic to Global cooling ratio under HiLLA-13. The regional temperature changes per degree global temperature change for all scenarios, models and regions are shown in Figure 5. The HiLLA-15 case has a similar ratio of Arctic to global mean cooling to the HiLLA-13 case, with ratios of 3.4, 3.0 and 2.5, for UKESM1, CESM2-WACCM and E3SMv3, respectively. In the Antarctic, there are larger model differences in

**Figure 2.** Global, Arctic (>66°N), Tropics (23°S-23°N) and Antarctic (>66°S) near-surface air temperature evolution under the background SSP2-4.5 scenario (grey) and the HILLA-13 and HILLA-15 scenarios (colours), in UKESM1 (a, d), CESM2-WACCM (b, e) and E3SMv3 (c, f). At the time of writing, the E3SMv3 SSP2-4.5 ensemble data was not available, hence the comparison to a single control run starting at 2035 in this case.

cooling (Figure 3). In particular, UKESM1 has a weak cooling response to both HiLLA-13 and HiLLA-15 in the Antarctic, of approximately 50% of the equivalent cooling in CESM2-WACCM. Neither UKESM1 nor CESM2-WACCM show an increase in Antarctic cooling under 15 km injection relative to 13 km, unlike for every other region assessed (and unlike E3SMv3).

Both UKESM1 and CESM2-WACCM have been used to simulate the ARISE-1.5K scenario, hereafter "ARISE", (Richter et al., 2022; Henry et al., 2024), which is useful for comparison of the regional cooling pattern under HiLLA scenarios to global feedback-controlled SAI scenarios. Figure 4 shows the result of this comparison, with the cooling in ARISE scaled linearly to the injection magnitude (12 Mt SO<sub>2</sub>/year) used under the HiLLA strategies. We see that efficiency of cooling is comparable to

185

**Figure 3.** Regional cooling efficiencies. For HiLLA cases, the change in near-surface air temperature in the annual and spatial mean is plotted for each region relative to the background simulation (SSP2-4.5 r1) over the final 20-years of simulation. For ARISE, the same quantity is used except that in this case, the cooling is scaled linearly to match the 12 Mt/year injection magnitude of the HiLLA cases, and the difference is taken as the ensemble mean of ARISE (5 members in both cases) against the 5 members of SSP2-4.5 from which these runs branch. Error bars for ARISE indicate the ensemble range across the 5 members. The 'Tropics' and 'Arctic' and are defined as (23°S-23°N) and >66°N, respectively.

or greater than ARISE over much of the high latitudes, particularly under the 15 km injection case. However, HiLLA-SAI is much less efficient than ARISE in the tropics and sub-tropics. With injection at 13 km, almost all regions equatorward of 40° see less than 40% of the ARISE efficiency in both UKESM1 and CESM2-WACCM. The average efficiency relative to ARISE over the tropics (23°S-23°N) is 28% in UKESM1 and 33% in CESM2-WACCM.

The reduced tropical cooling under HiLLA-SAI represents a combination of, first, reduced global mean cooling efficiency and, second, a more polar cooling profile. Of these, the former effect is larger. Tropical cooling per unit global cooling is approximately two thirds as large as the equivalent warming ratio under SSP2-4.5 (Figure 5) for HiLLA-13 (specifically, it is 75%, 61% and 64% of this value for UKESM1, CESM2-WACCM and E3SMv3, respectively). Whereas the global mean cooling efficiencies under HiLLA-13 in UKESM1 and CESM2-WACCM are 40% and 52% of their respective values under ARISE. The reduced tropical cooling efficiency under HiLLA-SAI is less pronounced if the injection altitude is increased to 15km in UKESM1 and E3SMv3, but not in CESM2-WACCM. Under HiLLA-15, the values for tropical cooling per unit

**Figure 4.** Spatial pattern of temperature change under HiLLA simulations. Panels (a) - (f) show the near-surface air temperature change relative to the background simulation, SSP2-4.5, over the final 20-years (2050-2069) of each HiLLA simulation. Panels (g) - (j) show the Ratio of cooling under HiLLA to that under ARISE-1.5K. 'Cooling' in both cases refers to the change in temperature relative to SSP2-4.5 over the period 2050-2069, and is normalized by the injection magnitude in each case. Panel (k) shows the zonal mean change in temperature relative to the 20-year period under SSP2-4.5 with the same temperature as that achieved under SAI over the period 2050-2069 (i.e. relative to the "target state"), normalized by the total global mean cooling under SAI.

global cooling are 92%, 62%, and 81% of each model's equivalent ratio under SSP2-4.5 warming, for UKESM1, CESM2-WACCM and E3SMv3, respectively. That is, in UKESM1, HiLLA-15 cools the tropics at 92% of the rate relative to the global mean cooling which would be needed to offset SSP2-4.5 warming, given sufficient injection magnitude. In CESM2-WACCM however, HiLLA-15 sees the same ratio of tropical to global cooling, as HiLLA-13, but increases the absolute tropical cooling due to the higher efficiency of global cooling per unit injection relative to HiLLA-13.

Given the strong seasonality of injection (and forcing), we also consider the seasonality of the temperature change under the
HiLLA scenarios. The SAI forcing peaks in local mid-/high-latitude summer in each hemisphere, so we might expect stronger
cooling in local summer than local winter, and a consequent reduction in both the local strength of the seasonal cycle outside
the tropics, and the absolute interhemispheric temperature difference. However, as shown in Figure A2, the simulated response

**Figure 5.** Regional temperature change per unit global mean temperature change, for the Arctic (>66°N), Tropics (23°S-23°N) and Antarctic (>66°S). Changes are relative to the background SSP2-4.5 scenario over [2050-2069] for the SAI scenarios. For SSP2-4.5 the temperature changes are [2050-2069] - [2015-2035] for UKESM1 and CESM2-WACCM. For E3SMv3, the SSP2-4.5 control run begins in 2035, and so we take the change [2060-2069] - [2035-2045].

of the seasonal cycle amplitude varies strongly by latitude. In the Arctic where faster warming in the winter than summer is decreasing the annual seasonal cycle, HiLLA-SAI acts to partially restore that greater seasonal cycle with cooling (due to the larger absolute cooling in winter, see Section 3.4). A similar effect is seen in the sub-polar Antarctic. However, there is a decrease in the strength of the seasonal cycle in the northern hemisphere mid-latitudes, and in the high southern latitudes, particularly in CESM2-WACCM. We do not plot this metric for E3SMv3 due to a lack of the needed historical simulation data.

Similarly, we can also assess the seasonality of the interhemispheric temperature difference, that is, the average Northern minus average Southern Hemisphere temperature, in a given month. We might expect summer-peaking forcing from SAI to reduce the amplitude of the seasonality in interhemispheric temperature difference, but the models do not simulate this (Figure A3). While the maximum (positive) interhemispheric temperature difference, during Boreal summer, is reduced, the minimum (largest negative value) during Austral summer is increased in magnitude (i.e. made more negative), such that the overall amplitude is within the range of the SSP2-4.5 ensemble. Put more simply, the absolute value of Northern Hemisphere cooling during its local winter is larger than the Southern Hemisphere cooling during its local summer, offsetting the reduction in seasonality which would otherwise be expected from the larger Northern Hemisphere cooling during its local summer. The implications of this for tropical precipitation are discussed in Section 3.6.

## 3.2 Aerosol distribution and optical depth

The seasonal cycle of zonal mean aerosol optical depth (AOD) distribution produced under the HiLLA-13 and HiLLA15 scenarios in each model is shown in Figure 6. This AOD pattern demonstrates that HiLLA-SAI produces a polar-peaking and highly seasonal aerosol burden and forcing. This seasonality is strongest in the 13 km case, which sees close to zero

**Figure 6.** Zonal mean total column AOD at 550nm. The difference relative to the background scenario, by month, averaged over the final 20 years of each simulation (2050-2069), is shown. Note that for CESM2-WACCM and E3SMv3, the 'AODVIS' variable is used, which is only outputted where insolation is non-zero, causing the data gaps in the polar winter. The green dotted lines show the injection locations and months.

H<sub>2</sub>SO<sub>4</sub> burden in the several months leading up to the deployment start in each hemisphere (Figures A4 and A5 show the full monthly evolutions of zonal H<sub>2</sub>SO<sub>4</sub> burden in UKESM1 and CESM2-WACCM). While the burden of sulfate is polar, there is also significant equatorward (and upward) transport (Figure 7). Considering the total sulfate burden in each latitude band (i.e. accounting for area) makes this clear. While the sulfate burden measured in mass per unit area peaks at the poles, the sulfate burden measured as mass per degree latitude peaks equatorward of injection, at 50-60° N/S (Figure 8). The majority of total additional atmospheric sulfate burden equatorward of the 60° N/S injection locations in both UKESM1 and E3SMv3 (60-75%, depending on scenario, model and hemisphere.

Following Zhang et al. (2024) we decompose the global cooling per unit injection into two components: the global mean AOD per unit injection, and the global mean cooling per unit global mean AOD. The resulting values for the three models under the HiLLA scenarios, along with ARISE for comparison where available, are shown in Figure 9. The AOD per unit

Figure 7. Zonal and seasonal mean burden of  $H_2SO_4$  mass-mixing ratio (kg per kg air), in UKESM1 (a-d) and CESM2-WACCM (e-h), averaged over the final 20 years of each simulation (2050-2069). Black triangles indicate the injection locations in each case. The top row (a, b) shows response under HiLLA-15, and the bottom row (c, d) under HiLLA-13 km. The left hand column (a, c) shows the Austral summer (Dec-Feb inclusive) burden, and the right column shows the Boreal summer burden (Jun-Aug inclusive). CESM2-WACCM burdens, which are outputted from the model in units of  $SO_4$ , were converted to  $H_2SO_4$  mass-mixing ratio for comparison with UKESM1. The monthly  $SO_4$  concentrations in E3SMv3 outputs were only recorded on individual isobaric surfaces and so are not included here.

injection is larger in UKESM1 than the other two models (panel a), but the reduced change in global temperature per unit AOD in UKESM1 (panel b) largely cancels this effect, so that the overall cooling per unit injection (panel c) is less varied between the three models. Compared to the ARISE scenario, the HiLLA scenarios result in more global cooling per unit AOD, by up to

Figure 8. Total additional burden of atmospheric SO<sub>4</sub> per degree latitude relative to SSP2-4.5, for the Southern Hemisphere (a), and Northern Hemisphere (b), in local summer in each case, under the HiLLA-13 and HiLLA-15 scenarios. Cumulative burdens, shown as a fraction of the hemispheric total, summing from the equator, are shown in (c) and (d). Note that no continuity is expected at the equator in (a) and (b), since different seasons are shown in the two subplots. The equivalent data output was not available in CESM2-WACCM, which is not shown here.

a factor of 2 (for HiLLA-13 in CESM2-WACCM). This greater cooling efficiency per unit AOD offsets part of the lower AOD per unit injection relative to ARISE, reducing the difference in global cooling efficiency between the scenarios. It is caused by the seasonality of the injection and the resulting AOD, which means the AOD produced coincides with greater insolation, combined with feedbacks in the polar regions (particularly the Arctic) which amplify the temperature response to the local forcing (e.g. Pithan and Mauritsen, 2014).

# 3.3 Sensitivity to injection strategy

In addition to the two central cases, for which we present 35-year simulations, we also simulate a set of nine 3-year simulations, from which we can assess the sensitivity of fast-responding systems to the injection latitude, altitude, longitude and seasonality.

**Figure 9.** Decomposition of cooling efficiency into (a) change in global mean AOD per 10Tg of SO<sub>2</sub> injection, (b) change in global mean temperature ('cooling') per 0.1 change in global mean AOD, and (c) change in global mean temperature per 10Tg injection. (c) is the product of (a) and (b). The HiLLA scenarios are compared against the ARISE-SAI-1.5 scenario in UKESM1 and CESM2-WACCM.

We focus our assessment on the response of aerosol optical depth to these features, because this stabilises at values within internal variability of the decadal mean under the 35-year simulations by the third year of injection.

The sensitivity of global mean AOD to altitude of injection, with injection latitude at 60°N/S, can be seen in the contrast between the HiLLA-13 and HiLLA-15 scenarios throughout this study. The three models show roughly a factor of two increase in mean AOD change in both hemispheres with the additional 2km of injection height (Figure 10). This difference is larger than the difference in SO4 burden between HiLLA-13 and HiLLA-15. For example, in UKESM1, the Northern Hemisphere AOD under HiLLA-15 is 202% of the HiLLA-13 value, but the total burden of SO4 is 136% of the HiLLA-13 value. The higher AOD per unit burden under HiLLA-15 in UKESM1 is likely due to the larger particle size under this strategy, which takes the effective radius closer to the optimal for scattering at around 300 nm (Dykema et al., 2016). The aerosol effective radius in UKESM1 and CESM2-WACCM under the two HiLLA scenarios is shown in Figure A6 (the data to produce the effective radius in E3SMv3 was not saved). The maximum effective radius under HiLLA-15 in UKESM1, approximately 250 nm, is around half the 500 nm maximum in CESM2-WACCM. The CESM2-WACCM values are similar to those reported by Bednarz et al. (2023a), for this model under their polar injection scenario.

The impact of varying injection latitude between 50 and 70°N/S on the resulting AOD is smaller than for variation in altitude, and is not consistent between the models (Figure 10). For example, at 15 km injection, moving the latitude from 60° to 50° increases the resulting AOD by 19% in UKESM1, leaves it unchanged in CESM2-WACCM, and decreases it by 29% in E3SMv3. Varying the injection longitude between of 0° and 180° (at 60° N/S and 13/15 km) has a smaller impact (Figure 11), although there may be a small gain in efficiency by injecting at 180° (i.e. over the Bering Sea rather than the Norwegian Sea in the Northern Hemisphere). Similarly, within the spring/early-summer injection regime, the influence of further fine-tuning of the seasonality of injection is generally small and is inconsistent across the models (Figure 11). In the Southern Hemisphere,

**Figure 10.** Change in total atmospheric column aerosol depth (AOD) from background scenario (SSP2-4.5) under injection at 15 km (a, b) and 13 km (c, d) and varying latitude. All values are for the third year of these (3-year) simulations. All runs here have longitude 180°E, and seasonality MAMJ/SOND. Values shown are hemispheric means for the (a,c) Northern Hemisphere and (b, d) Southern Hemisphere. The uncertainty range shown for the 60° injection cases is one standard deviation of the year-to-year variability, derived from the 35-year HiLLA experiments.

AOD is higher for September-December (SOND) injection in UKESM1, but in the other models, variation is within the year-to-year variability (one standard deviation) of the SOND value. In the Northern Hemisphere, where AOD differences between the models are larger, the longer injection season of February-June (FMAMJ) results in higher mean AOD in UKESM1 and E3SMv3, but March-June (MAMJ) is marginally higher in CESM2-WACCM.

#### 3.4 Arctic Amplification and the Seasonality of Arctic Cooling

Given the strong amplification of Arctic warming with its marked Autumn/Winter peak (e.g. Rantanen et al., 2022), it is of interest to examine the seasonality of the Arctic cooling response to HiLLA-SAI. The local Arctic forcing under SAI drops to zero in the Winter months when there is no insolation – or can be positive if injecting throughout the year due to longwave absorption by the sulfate aerosols (Lee et al., 2021; Duffey et al., 2025a) – but the late autumn and winter months (particularly November and December) see the largest absolute Arctic temperature change under both HiLLA-SAI and ARISE (Figure

**Figure 11.** Change in total atmospheric column aerosol depth (AOD) from background scenario (SSP2-4.5) under varying longitude (a, b) and seasonality (c, d). Values shown are hemispheric means for the (a, c) Northern Hemisphere and (b, d) Southern Hemisphere. The uncertainty range shown for the 180°E cases is one standard deviation of the year-to-year variability, derived from the 35-year HiLLA-SAI experiments. Panels (a) and (b) show the impact of varying longitude of injection from 180°E (the default in our HiLLA scenarios) to 0°E, under injection at 13 and 15 km. All values are for the third year of these short (3-year) sensitivity simulations. All runs in (a) and (b) have latitude of injection at 60°N/S and seasonality MAMJ/SOND. Panels (c) and (d) show the impact of varying the precise seasonality of injection relative to the central case (March-June, MAMJ). All runs in panels (c) and (d) use injection at 60°N/S, 180°E, 13 km. No E3SMv3 data is available for the MAMJJ/SONDJ case, due to a mistake in injection settings for this simulation.

12). This result concurs with previous simulations (e.g. Lee et al., 2023a). It suggests that the same drivers which cause Arctic Amplification of warming to peak in the Winter – including the lapse rate feedback and ocean heat fluxes (Pithan and Mauritsen, 2014; Hahn et al., 2021; Duffey et al., 2025b) – also apply for cooling under SAI.

While the absolute cooling under HiLLA-SAI peaks in Winter, this peak is weaker than the peak in winter-time warming. As such, HiLLA-SAI overcools the Arctic summer, and undercools the winter, in the sense that if HiLLA-SAI was deployed to offset a given annual mean Arctic warming, there would be positive residual warming in winter and negative residual warming in summer (Figure 12, right column). That is, HiLLA-SAI is reducing the strength of the Arctic temperature seasonal cycle relative to a baseline with equal annual mean temperature, even though it increases the seasonal cycle relative to the background warming scenario (Figure A2). This finding was previously shown for the Geoengineering Large Ensemble Project (GLENS) simulations by Jiang et al. (2019) and also applies for the ARISE scenario in UKESM1 and CESM2-WACCM (Figure 12).

**Figure 12.** Seasonality of Arctic warming and SAI cooling. The left hand side plots (a, c, e) show Arctic (>66°N) warming (SSP2-4.5 2050-2069 - pre-industrial), and cooling (SAI scenario - SSP2-4.5, 2050-2069), for the three models. Values are normalized by the annual mean change. The right column plots (b, d, f) show the residuals (warming + cooling) from the left side plots.

The seasonal temperature residuals shown in Figure 12 are, in almost all months, larger under HiLLA-SAI than under ARISE, meaning this reduction in seasonality relative to the target state is larger under the more seasonal and polar forcing of the HiLLA strategy.

## 3.5 Sea-ice response

Local temperature is a strong control on total Arctic and Antarctic sea-ice cover (e.g. Ridley et al., 2012) and all SAI simulations show increases in both Arctic and Antarctic sea-ice area relative to the background warming scenario (Kravitz et al., 2013; Berdahl et al., 2014; McCusker et al., 2015; Jones et al., 2018; Jiang et al., 2019; Lee et al., 2021, 2023a; Goddard et al., 2023). As such, it is not surprising to see that sea-ice area is increased in both the winter maximum and summer minimum in both hemispheres and in all models under HiLLA-SAI (Figure 13). All three models share similar trends in sea ice area response to SAI. In the Arctic, the September minimum sea ice is increased relative to SSP2-4.5 as is the March maximum. Generally, the increases are greater for the HiLLA-15 cases than HiLLA-13. However, UKESM1 and CESM2-WACCM have much stronger increases in September sea ice than E3SMv3. UKESM1 shows a dramatic initial increase in September sea ice to the first decade of SAI which declines steeply starting in 2045 to nearly ice-free by 2070. CESM2-WACCM shows a similarly strong initial response but has a much shallower decline in September sea ice area through 2070. CESM2-WACCM,

sees an ice-free Arctic by the 2040s under SSP2-4.5, but has around 4 Mkm<sup>2</sup> of September sea-ice area under HiLLA-15 and around 3 Mkm<sup>2</sup>, approximately the observed 2024 value (Fetterer et al., 2025), under HiLLA-13 in the 2060s. E3SMv3 has a much smaller increase in September sea ice area, which remains near or below ice free under both SSP2-4.5 and SAI. The impact on the Arctic March maximum area is smaller across the models, particularly in CESM2-WACM and E3SMv3, for which the background decline in the winter maximum under SSP2-4.5 is small.

For Antarctic sea ice, all models show an increase in February and September Antarctic sea ice response to SAI. Unlike for Arctic sea ice, E3SMv3 shows a similar magnitude increase in sea ice area to UKESM1 and CESM2-WACCM with SAI. As with the temperature impacts, and unlike the Arctic response, there is little difference in the Antarctic February sea-ice minimum response to HiLLA-13 versus HiLLA-15. Both strategies increase the Antarctic February minimum sea-ice area in all models, with the response in CESM2-WACCM (of  $\sim$ 2.5 Mkm $^2$ ) around twice the size of that in the other two models, but no model shows a clear increase in the response for the 15 km injection case relative to the 13 km. However, the additional injection height produces a stronger sea-ice increase for the Antarctic September sea ice maximum.

Temperature is the principal driver of Arctic sea ice change (Notz and Stroeve, 2018), and CMIP6 models consistently show a linear decline in annual minimum sea ice area with temperature (Notz and Community, 2020). In our HiLLA simulations, Arctic annual mean temperature is a good predictor of the sea ice area at both annual minimum and maximum, suggesting that the relationship between local temperature and pan-Arctic sea ice cover which applies under the background SSP2-4.5 scenario also holds under SAI (Figure A7). The Antarctic sea ice area is plotted against local temperature in Figure A8. In the Antarctic case, atmospheric circulation is a stronger control on interannual variability of sea ice (e.g. Blanchard-Wrigglesworth et al., 2021) and the relationship is weaker.

# 310 3.6 Precipitation

The global precipitation response under HiLLA-15 is shown in Figure 14 (see Figure A9 for the 13 km injection case). Our simulations are not sufficient in duration or number of ensemble members to present a detailed assessment of the impact on regional precipitation given its large internal variability. With our 20-year assessment for one ensemble member, most regional changes do not meet a 5% false discovery rate significance threshold (Wilks, 2016), particularly at sub-annual timescales. However, a general reduction in precipitation relative to the background (warmer) SSP2-4.5 scenario is evident, particularly in the high latitudes, where the changes are significant in the annual mean in all three models across most of the Arctic and Antarctic. This reduction in high-latitude precipitation relative to the control is expected given the strong cooling in these regions, and given the increase in precipitation with warming which is simulated under SSP2-4.5 for the Arctic (e.g. McCrystall et al., 2021) and Antarctic (e.g. Bracegirdle et al., 2020).

All three models show at least some regions of significant change to annual mean precipitation in the tropics. One potential impact of HiLLA-SAI which we seek to check for here is a reduction in local summer rainfall in low latitude regions on either side of the equator, associated with a reduced amplitude of Intertropical Convergence Zone (ITCZ) seasonal migration due to stronger cooling in the summer hemisphere. By analogy to the ITCZ shifts associated with asymmetric SAI forcing (Haywood et al., 2013), such an effect might be hypothesised given the seasonal migration of the ITCZ towards the warmer hemisphere,

**Figure 13.** Sea-ice area under HiLLA-SAI simulations and SSP2-4.5, by region and minimum/maximum months, for each model. The shaded region for SSP2-4.5 is the ensemble range, the solid line is the ensemble mean, and the dashed line is the first member (from which the HiLLA scenarios branch).

combined with the HiLLA scenarios' summer hemisphere-only AOD pattern. No clear evidence of this effect is seen in the rainfall changes across the three models, although the significant drying during boreal summer in central Africa in UKESM1 and CESM2-WACCM could be suggestive of this mechanism. However, this is only assessed from a short timeseries, which is inadequate to detect anything but very large precipitation changes given the internal variability. As such, tropical precipitation changes under HiLLA scenarios will require assessment in longer scenarios controlling for global temperature, and with more ensemble members, before any firm conclusions can be drawn.

**Figure 14.** Change in precipitation under HiLLA-15 relative to SSP2-4.5 over the period 2050-2069. Maps (a-i) show grid cell changes, over the 20 years of assessment, relative to the ensemble mean of the SSP2-4.5 simulation (10 members each for UKESM1 and CESM2-WACCM, but only 1 member for E3SM). Stippling marks changes which are significant at a 5% False Discovery Rate, following Wilks (2016). The single SSP2-4.5 member for E3SMv3 means a larger signal is required for significance in this case. Zonal mean plots are also shown (j-l), as a percentage change from the SSP2-4.5 ensemble mean. For UKESM1 and CESM2\_WACCM, the 10-member ensemble range in variation of the 20-year mean is shaded.

# 3.7 Other impacts: sulfur deposition and stratospheric ozone

Once the overall increased stratospheric aerosol burden is stable, the global additional magnitude of sulfur deposition is equal to the injected mass. The global cooling efficiency reduction for HiLLA-SAI compared to high-altitude strategies described above therefore implies an increase in overall deposition per unit of cooling of a factor of 2-2.5 for HiLLA-13, relative to high-altitude low-latitude injection strategies such as ARISE. If we compare against ARISE on a per unit injection basis, the total deposition must be the same, but the regional patterns can differ. Figure A10 shows the zonal mean SO<sub>4</sub> deposition in UKESM1 and CESM2-WACCM (data to support this analysis was not saved in the E3SMv3 simulations). Overall, the zonal patterns

under HiLLA-SAI are similar to those under ARISE, although with greater mid/high-latitude Northern Hemisphere deposition (and less mid/high-latitude Southern Hemisphere deposition) in CESM2-WACCM than ARISE (which injects mostly in the Southern Hemisphere in this model), and with somewhat higher sub-polar Southern Hemisphere deposition in UKESM1. Maps of the change in deposition relative to the background SSP2-4.5 scenario are shown in Figure A11. We see from these maps that, first, the largest absolute deposition values occur equatorward of injection, but, second, that the largest changes relative to the local background occur over more pristine regions in the high latitudes and the Southern Ocean, particularly Greenland and the Canadian Arctic. This result has also been highlighted under lower-latitude injections (Visioni et al., 2020), in which case the stratospheric circulation moves aerosols towards the mid-to-high latitudes and results in increased deposition in the same areas.

Changes in stratospheric ozone concentrations relative to the SSP2-4.5 background simulation under the HiLLA scenarios are shown in Figure A12. SAI impacts ozone through changes to both chemistry and dynamics (Tilmes et al., 2018, 2021; Bednarz et al., 2023a). Of the three models, only CESM2-WACCM includes heterogeneous chemistry in the stratosphere, and so only this model is expected to capture the depletion due to increased heterogeneous reactions on sulfate aerosols. In this model, decreases of ozone concentration of up to around 30% are found in the Antarctic stratosphere, and significant reductions are also seen in the equatorial lower stratosphere and the Arctic. In the other two models, where only dynamic changes are captured, there are significant increases in ozone concentration in the tropical lower stratosphere, and, in UKESM1, in both polar regions.

## 4 Discussion and conclusions

In this study, we have presented a first multimodel assessment of HiLLA-SAI, to characterise the global climate response to this form of SAI deployment. HiLLA-15, that is, an annual injection of 6 Mt SO<sub>2</sub> per hemisphere during the local spring and early summer in the subpolar lower stratosphere (at 15 km and 60°N/S) is simulated to reduce global mean temperature by 0.9 - 1.0 °C across the three models. Alongside this global cooling, the models show strong polar cooling, an increase in sea-ice area in both hemispheres, a decrease in high-latitude precipitation and an increase in the amplitude of the Arctic seasonal cycle resulting from the greater absolute cooling during the winter months. Our modelling therefore suggests that HiLLA deployment could be an effective means of reducing many of the rapid and consequential changes occurring in the Arctic and Antarctic. At the same time, HiLLA-SAI would be a global, not a polar, climate intervention. It would produce cooling, precipitation changes and other regional climate impacts, as well as sulfur deposition, over the entire globe. The majority of the total increase in atmospheric burden is modeled to be equatorward of the 60°N/S injection locations.

We also compared our HiLLA simulations in UKESM1 and CESM2-WACCM against conventional high-altitude low-latitude SAI, which we characterise using the ARISE simulations (Richter et al., 2022). HiLLA-15 achieves approximately equal Arctic cooling efficiency as the ARISE scenario. However, there is strongly reduced cooling efficiency in the low latitudes in HiLLA-SAI relative to conventional high-altitude low-latitude strategies. Under HiLLA-13, the Tropics mean cooling is less than 0.4°C for 12 Mt annual injection in all three models, a local cooling efficiency of around 25% of the ARISE strategy

in UKESM1 and CESM2-WACCM. This reduction represents a combination of reduced global mean cooling efficiency and a more polar cooling profile, and the former effect is larger. While the HiLLA strategies are more polar than high-altitude subtropical SAI, this effect can be overstated. Across the three models, tropical cooling per unit global cooling under HiLLA-13 is at least 61% of the equivalent ratio under GHG warming, and this ratio increases under 15km injection in two of three models. Our simulations suggest that HiLLA-SAI, and therefore SAI using modified existing aircraft rather than newly-developed high-altitude aircraft, is a plausible early-stage deployment option for both polar and global climate intervention strategies. While both 13 km and 15 km injection see meaningful impacts, the higher altitude raises efficiency markedly. The three models

While both 13 km and 15 km injection see meaningful impacts, the higher altitude raises efficiency markedly. The three models suggest close to 1°C more cooling per 12 Mt SO<sub>2</sub> could be achieved in the Arctic, 0.75 °C more cooling in the Antarctic, and around 50% greater global mean cooling, under 15 km injection than 13 km. This sensitivity to altitude is important given that 13 km is close to the certified altitude ceiling for large jetliners such as the Boeing 777 which could be used to deliver large payloads with relatively minor modifications (Smith et al., 2024). Our results suggest that using aircraft capable of deploying several km higher than this would strongly increase efficiency for HiLLA strategies.

Varying the latitude of injection between 50° and 70° N/S can make a large difference to resulting hemispheric mean AOD, particularly in the Southern hemisphere. In E3SMv3, injecting at 60°S rather than 50°S at 13 km altitude increases the resulting AOD by around a factor of two. While less extreme in the other two models, all three show an increase in AOD for injection at 60°S relative to 50°S, at 13 km, suggesting that if altitude is limited to this height, then injecting at higher Southern latitudes than the most Southerly airports in Patagonia, at 53-55°S (Smith et al., 2024), would increase efficacy. In the Northern Hemisphere, there is no equivalent increase in AOD with increased latitude from 50° to 60° at 13 km altitude. At the higher altitude of 15 km, there is no consensus between the models on the impacts of that same increase in latitude - in both hemispheres shifting from 50° to 60° increases the resulting AOD E3SMv3 but decreases it in UKESM1, and has no impact in CESM2-WACCM. In general, our results suggest there is limited sensitivity to the longitude and precise seasonality (within the spring/early summer regime) of injection under HiLLA strategies.

We have made only a preliminary assessment of the impacts of HiLLA-SAI on changes in sulfur deposition, stratospheric ozone, and changes to the impacts on tropical precipitation arising from the polar and seasonal nature of the SAI forcing. There may well be further shortcomings to HiLLA-SAI strategies which our limited simulations here cannot identify. Exploring the processes which drive such differences, and producing a comprehensive assessment of the trade-offs between HiLLA and other SAI strategies, is an important near-term goal for the SAI research community. Based on the sensitivity analysis presented in this paper, a new experiment was designed and at present new simulations under the GeoMIP project, "G6-1.5k-HiLLA", are ongoing in the three models used here (UKESM1, CESM2-WACCM and E3SMv3) to support this research effort (Visioni et al., 2025). These simulations use seasonal injection at 15 km, 60° N/S, exactly as in HiLLA-15, except that the overall magnitude of injection is varied to control global mean temperature response to the 2020-2039 average temperature in each model. Since this target is the same as that for the G6-1.5K-SAI scenario (Visioni et al., 2024), these new HiLLA simulations will allow a first comparison of HiLLA versus global high altitude SAI simulations targeting the same temperature outcomes in a multi-model setting.

Code and data availability. All plotting and analysis code is available at https://github.com/alistairduffey/HiLLA\_MM and is archived on Zenodo (Duffey, 2025). Data needed to reproduce the analysis is archived on Zenodo (Duffey et al., 2025c). The version of CESM used in this experiment, version 2.1.5, is freely available for download; links to instructions and documentation are available at https://www.cesm.ucar. edu/models/cesm2. E3SMv3 model code can be accessed at https://github.com/E3SMv3-Project/E3SMv3. Additional model documentation is available at: https://docs.E3SMv3.org/E3SMv3/index.html.

# 10 Appendix A: Supplementary Figures

**Figure A1.** Annual and global mean change in mean near-surface air temperature relative to the background scenario SSP2-4.5, for HiLLA-13 and HiLLA-15 scenarios, in UKESM1, CESM2-WACCM and E3SMv3.

**Figure A2.** Annual amplitude of the (monthly mean) seasonal cycle, in the zonal mean, for UKESM1 and CESM2-WACCM, SSP2-4.5 (2050-2069) and HiLLA-SAI scenarios (2050-2069). In both cases the amplitude plotted is the anomaly relative to the historical simulation (1995-2015) period. E3SMv3 is not plotted due to lack of simulation availability for the baseline period.

**Figure A3.** Seasonal cycle of interhemispheric temperature difference. The figure shows the mean (a-c), maximum (d-f), minimum (g-i) and annual amplitude (max - min, j-l) of the interhemispheric temperature difference, defined as the northern hemisphere mean temperature minus the southern hemisphere mean temperature. The seasonal cycle is calculated on the monthly resolution data. The shaded region is the ensemble range over 5 members, for the SSP2-4.5 simulations in UKESM1 and CESM2-WACCM, and the colored lines show the evolution under the HiLLA-15 SAI scenario. The HiLLA-13 scenario is not shown, but has a qualitatively similar pattern.

Figure A4. UKESM1 zonal mean monthly burden of  $H_2SO_4$ , as a mass-mixing ratio (kg per kg air), averaged over the final 20 years of each simulation (2050-2069). Black triangles indicate the injection locations in each case. The top row (a-l) shows response under HiLLA-15, and the bottom row (m-x) under HiLLA-13.

Figure A5. CESM2-WACCM zonal mean monthly burden of  $H_2SO_4$ , as a mass-mixing ratio (kg per kg air), averaged over the final 20 years of each simulation (2050-2069). Black triangles indicate the injection locations in each case. The top row (a-l) shows response under HiLLA-15, and the bottom row (m-x) under HiLLA-13.

Figure A6. Effective radius of sulfate aerosols in UKESM1 (top) and CESM2-WACCM (bottom) under HiLLA-13 and HiLLA-15, for each season. The mass mixing ratio  $(10^{-8}\text{kg/kg air})$  is shown in the contours. All values are the 20-year means over the final 20-years of the simulations (2050-2069). The effective radius for UKESM1, where it is not a direct model output, is calculated as in Visioni et al. (2023). The required geometric standard deviations for each of the four aerosol modes in UKESM1 are given in Walters et al. (2019), and we use a sulfate density of 1769 kg m<sup>-3</sup> following Mann et al. (2010).

**Figure A7.** Arctic sea ice area by Month and model, over the HiLLA-SAI and SSP2-4.5 scenarios, plotted against the annual mean Arctic (>66°N) temperature.

**Figure A8.** Antarctic sea ice area by Month and model, over the HiLLA-SAI and SSP2-4.5 scenarios, plotted against the annual mean Antarctic (>66°S) temperature.

Figure A9. As in Figure 14, except showing the HiLLA-13 scenario rather than the 15 km case.

**Figure A10.** Zonal mean deposition of SO<sub>4</sub>, in grams of sulfur, under HiLLA-SAI in UKESM1 and CESM2-WACCM, along with the ARISE deposition, scaled linearly to the same annual injection magnitude (12Mt) as the HiLLA cases. All lines show the 2050-2069 mean except the 'SSP2-4.5 (2020s)' line which shows the present day (2020-2029) simulated value.

 $\textbf{Figure A11.} \ Deposition \ of \ SO_4 \ under \ HiLLA-SAI \ in \ UKESM1 \ and \ CESM2-WACCM \ , in absolute units \ (milligrams \ S \ per \ m2 \ per \ year \ (left) \\ and \ as \ a \ percentage \ of \ the \ local \ background \ deposition \ under \ SSP2-4.5 \ (right). \ All \ values \ are \ time \ means \ over \ the \ 20-year \ period \ 2050-2069.$ 

**Figure A12.** Ozone response - percentage change in ozone concentration relative to the SSP2-4.5 background over the final 20-years of simulations (2050-2069). The gray contours show the background concentration in parts per million. The stippling marks changes which are significant at a 5% False Discovery Rate, following Wilks (2016), using three ensemble members from the SSP2-4.5 scenario for UKESM1 and CESM2-WACCM, but only the single available member for E3SMv3, meaning a larger signal is required for significance in this case. Only CESM2-WACCM has stratospheric heterogeneous chemistry active. The tropopause height in each model is shown as the black dashed line.

420

Author contributions. AD ran the UKESM1 simulations, performed the analysis and prepared the original manuscript. WL ran the CESM2-WACCM simulations. LW ran the E3SMv3 simulations, with assistance from BW. PI and MT supervised the project. DGM contributed to conceptualization and methodology, along with MH, DV and PI. All authors contributed to review and editing of the manuscript.

Competing interests. The authors declare that no competing interests are present.

415 Acknowledgements. AD's contribution was funded by the Natural Environment Research Council (NERC) London Doctoral Training Partnership (DTP) Grant NE/S007229/1.

This work was partly supported by the Laboratory Directed Research and Development program at Sandia National Laboratories. Sandia National Laboratories is a multi-mission laboratory managed and operated by National Technology & Engineering Solutions of Sandia, LLC (NTESS), a wholly owned subsidiary of Honeywell International Inc., for the U.S. Department of Energy's National Nuclear Security Administration (DOE/NNSA) under contract DE-NA0003525. This written work is coauthored by an employee of NTESS. The employee, not NTESS, owns the right, title and interest in and to the written work and is responsible for its contents. Any subjective views or opinions that might be expressed in the written work do not necessarily represent the views of the U.S. Government. The publisher acknowledges that the U.S. Government retains a non-exclusive, paid-up, irrevocable, world-wide license to publish or reproduce the published form of this written work or allow others to do so, for U.S. Government purposes. The DOE will provide public access to results of federally sponsored research in accordance with the DOE Public Access Plan. MH is funded by SilverLining through the Safe Climate Research Initiative.

The CESM project is supported primarily by the National Science Foundation. Computational support and computer and data storage services, including the Derecho supercomputer (doi: 10.5065/qx9a-pg09), were provided by the Computational and Information Systems Laboratory (CISL) at NSF NCAR.

Support for WRL has been provided in part by the Quadrature Climate Foundation, Grant No. 01-21-000349.

Support for DGM and DV was provided in part by the Quadrature Climate Foundation.

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
