# Peer review of "The global climate response to High-Latitude Low-Altitude Stratospheric Aerosol Injection (HiLLA-SAI)"

_EGUsphere, 2025_

## Referee Comment (RC2)

**Summary:**

This paper presents the first multi-model comparison of Earth-system model simulations of high-latitude low-altitude (HiLLA) stratospheric aerosol injection (SAI) using CESM2-WACCM, UKESM1, and E3SMv3. HiLLA SAI is proposed as a potential method of climate intervention which could serve as an early-stage deployment option since existing aircraft could be repurposed for deployment, in contrast to SAI deployment methods which consider lower latitude and higher altitude injection and require the development of new aircraft. The results of this manuscript assess the global and regional responses of temperature, precipitation, aerosol optical depth, sulfate burden, sea ice, sulfur deposition, and stratospheric ozone to idealized 35-year HiLLA SAI deployment simulations which inject 12 Tg $SO_2$ per year into the polar stratosphere, with injections occurring at 60ºN from March through June, and at 60ºS from September to December. Sensitivity to injection height (13 vs. 15 km in altitude), altitude, seasonality, and longitude are considered. The main finding of this work is that HiLLA SAI deployment could reduce global temperatures. Additionally, HiLLA SAI deployment is found to induce strong polar cooling, increased sea-ice area, decreased high-latitude precipitation, and an increased Arctic seasonal cycle, relative to the background climate change scenario and to a low-latitude high-altitude SAI deployment scenario. However, the cooling efficiency, defined as the magnitude of near-surface air temperature change per unit of injected $SO_2$, is reduced in HiLLA-SAI scenarios compared to traditional high altitude injection strategies.

The simulations and key results presented in this study contribute to a growing body of literature evaluating possible SAI deployment strategies. The analysis of the global and regional climate impacts of the HiLLA SAI simulations, as well as the sensitivity testing of HiLLA SAI to injection seasonality, latitude, longitude, and altitude are overall comprehensive and well done. However, there are a few major comments and several minor comments that should be addressed prior to publication.

**Major Comments:**

The authors should consider adding in some more commentary on some of the major differences between the considered models (CESM2-WACCM, UKESM1, E3SMv3), particularly when relevant to inter-model differences in the climate response to HiLLA deployment.

-   One example when this is already done nicely is in Section 3.7, with the discussion of changes in stratospheric ozone, CESM2-WACCM is the only model that experiences decreases in stratospheric ozone in response to HiLLA SAI deployment since this model includes heterogeneous chemistry in the stratosphere, whereas UKESM1 and E3SMv3 do not.
-   Lines 223-233 discuss AOD per unit injection and global mean cooling per unit global mean AOD. Why might UKESM1 differ from CESM2-WACCM and E3SMv3 in AOD per 10 Tg injection and change in global mean temperature per 0.1 AOD?

- Lines 250-260 assess the impact of varying injection latitude, longitude, and seasonality. Why might there be so much inconsistency between considered models, particularly when considering the latitude of injection?

The discussion of Figures 7 and 8 should be considered carefully. Figure 7 shows only results from UKESM1 and CESM2-WACCM. Figure 8 shows only results from UKESM1 and E3SMv3. This should probably be noted in the text. Why might the majority of the sulfate be located equatorward of the injection location?

In section 3.6, statistical significance testing is discussed relative to a "5% false discovery rate significance threshold (Wilks, 2016)." The methods outlined in Wilks (2016) for adjusting for the False Discovery Rate could be applied to a variety of different statistical tests (e.g., student's t-test). As such, the authors should include more details on the specific statistical test where p-values were then adjusted to account for the False Discovery Rate. Additionally, the authors should elaborate on why they chose to test for significance for precipitation changes, but not for other variables like temperature and AOD, for example. The authors might consider what value significance testing brings to analysis of precipitation changes, and whether the interpretation of the analysis would be altered if it was removed, given the relatively sparse regions of significance, and considering that only a single ensemble member was conducted for each simulation. Perhaps plotting precipitation as a percent change plot relative to SSP2-4.5 or to historical data would be more informative.

In the Discussion and Conclusions section, please add a bit more discussion on the limitations of the results presented in this study. An important example would be that there was only a single simulation run for each model. Another might be that these simulations considered HiLLA SAI in the context of one emissions scenario and only presented one idealized deployment strategy. Also in this section, consider a bit of discussion on the potential implications of injecting only in one hemisphere at a time seasonally, and what this might mean for global circulation and thus, regional climate. Given the asymmetry of this deployment strategy, this will be an important aspect of risk assessment.

There are some components of the figures that are not well described in the caption or in the text. Please make sure that the figure captions are completely descriptive of what is shown in each figure. Here is a non-exhaustive list of changes that should be made:
- "r1" is used to represent the first ensemble member of the SSP2-4.5 simulation in Figure 2 and Figure 3. This should be written out explicitly in the figure caption.
- Figure 2: This is only briefly mentioned in the manuscript, and nothing about the results shown in the figure are mentioned. The authors should add a brief comment on what is shown in the figure (e.g., what is the point of having the figure in the manuscript at all?) or consider moving the figure into the supplement.

- Figure 2 caption: Mention that for UKESM1 and CESM2-WACCM, the solid gray line represents the ensemble mean under SSP2-4.5, the dashed line represents the 'r1' ensemble member, and that the light gray shading is the spread of the ensembles (that is what I assumed it to be at least). Additionally, for the HiLLA lines, mention that HiLLA-13 is represented by the solid colored lines and HiLLA-15 is represented by the dash-dotted lines.
- Consider splitting Figure 4 into two where one figure includes (a)-(f) and the other includes (g)-(k).
- Figure A3: Even if HiLLA-13 has a "qualitatively similar pattern", it would be informative to also include the data from this simulation in the plot.
- Figure A10: What is shown by the blue line in plot (b) showing the zonal mean deposition of $SO_4$ in CESM2-WACCM? Why aren't the zonal means of HiLLA-13 and HiLLA-15 simulations shown for this plot?

**Minor Comments:**
- The authors should comb through the manuscript, figure titles, and figure captions to ensure that there is consistent labeling of models and simulations throughout. There are several different abbreviations used for the models where CESM2-WACCM is also noted as "CESM" and "CESM2", UKESM1 is also noted as "UKESM", and E3SMv3 is noted as "E3SM."
- Line 12: "For 13 km inject" → "For 13 km **injection**"
- Lines 22-25: "cooling efficiency" should be defined here, where it is first mentioned, rather than in Lines 41-42 where it is currently defined.
- Lines 43-44: Please describe briefly what is meant by "a less favorable background circulation."
- Line 55: Please indicate what some of these local feedbacks are (e.g., ice-albedo), they do not need to be comprehensively explained.
- Line 62: "Beyond the overall global cooling efficiency…" → "Beyond the overall **reduced** global cooling efficiency…"
- Lines 87-88: What is the specific version of UKESM1 used for these simulations?
- Lines 122-123: "... ECMWF Reanalysis version 5 (ERA5) reanalysis…" → "... ECMWF Reanalysis version 5 (ERA5)..."
- Line 125: Might be worth mentioning that SSP2-4.5 is a moderate emissions scenario in line with current policies and then can cite O'Neill et al. (2017).
- Line 140: "we vary (independently)" → "we independently vary"
- Lines 159-160: "Raising the altitude of injection to 15 km increases the global cooling efficiency by 60%, 34%, and 62% in UKESM1, CESM2-WACCM, and E3SMv3, respectively." I would recommend changing the percent changes to the actual cooling efficiencies of the 15 km injection.

- Lines 161-162: As mentioned in the comment for Line 55, consider explicitly mentioning a few of the feedbacks which drive Arctic Amplification.
- Lines 163-164: Reference Figure 5.
- Lines 177-178: "We see that efficiency of cooling is comparable to or greater than ARISE over much of the high latitudes, particularly under the 15 km injection case." Rephrase this sentence. It is a bit misleading to say that cooling efficiency is "comparable or greater" than ARISE for HiLLA SAI. This is particularly so for the HiLLA-13 case. Perhaps it could be something like, "Cooling efficiency is lower compared to ARISE over most regions of the globe for HiLLA-13. For the 15 km injection case, however, there are large regions of the globe, particularly over high latitudes, where cooling efficiency is comparable or greater than ARISE."
- Lines 217-218: Consider rephrasing this sentence to something like, "While the largest burden of sulfate occurs at the poles, there is also transport equatorward and upward." Take care of use of the word "significant."
- Line 221: "... additional atmospheric sulfate burden equatorward…" → "... additional atmospheric sulfate burden **is** equatorward…"
- Line 236: What is meant by "fast-responding" systems?
- Line 243: Where is the change in total SO4 burden shown? Reference Figure 8 here.
- Lines 281: Change "all SAI simulations" to "all **previously** conducted SAI simulations" or something similar
- Line 320: This sentence is a bit misleading. The regions of statistically significant changes in tropical precipitation are relatively small, particularly for CESM2 and E3SMv3. Consider removing this sentence or rewording it.
- Lines 357-361: Add a brief statement mentioning how the HiLLA-13 impacts compare to HiLLA-15.
- Lines 372-373: What is meant by "this effect can be overstated"?
- Line 377: Explain why higher altitude of injection leads to higher cooling efficiency.
- Figure 1 caption: Remove the extra "for"
- Figure 4: A (k) label is missing from panel (k) plot.
- Figure 9 caption: "10Tg" → "10 Tg"
- Figure 12 caption: Include the definition of Arctic when it is first mentioned.
- Figure 13 caption: The regions and their latitude definitions should be explicitly defined in the figure caption.

**References**

O'Neill, B. C., Kriegler, E., Ebi, K. L., Kemp-Benedict, E., Riahi, K., Rothman, D. S., et al. (2017). The roads ahead: Narratives for shared socioeconomic pathways describing world futures in the 21st century. *Global Environmental Change, 42*, 169–180. https://doi.org/10.1016/j.gloenvcha.2015.01.004

Wilks, D. S. (2016). "The Stippling Shows Statistically Significant Grid Points": How Research Results are Routinely Overstated and Overinterpreted, and What to Do about It. https://doi.org/10.1175/BAMS-D-15-00267.1